# Effect of High Fat and Fructo-Oligosaccharide Consumption on Immunoglobulin A in Saliva and Salivary Glands in Rats

**DOI:** 10.3390/nu13041252

**Published:** 2021-04-10

**Authors:** Yuko Yamamoto, Toshiya Morozumi, Toru Takahashi, Juri Saruta, Wakako Sakaguchi, Masahiro To, Nobuhisa Kubota, Tomoko Shimizu, Yohei Kamata, Akira Kawata, Keiichi Tsukinoki

**Affiliations:** 1Department of Dental Hygiene, Kanagawa Dental University, Junior College, 82 Inaoka, Yokosuka, Kanagawa 2388580, Japan; yamamoto.yuko@kdu.ac.jp; 2Division of Periodontology, Department of Oral Interdisciplinary Medicine, Graduate School of Dentistry, Kanagawa Dental University, 82 Inaoka, Yokosuka, Kanagawa 2388580, Japan; morozumi@kdu.ac.jp; 3Department of Health and Nutrition, Faculty of Human Health, Kanazawa Gakuin University, 10 Sue-machi, Kanazawa, Ishikawa 9201392, Japan; t-takahasi@kanazawa-gu.ac.jp; 4Division of Environmental Pathology, Department of Oral Science, Graduate School of Dentistry, Kanagawa Dental University, 82 Inaoka, Yokosuka, Kanagawa 2388580, Japan; saruta@kdu.ac.jp (J.S.); sakaguchi@kdu.ac.jp (W.S.); n.kubota@kdu.ac.jp (N.K.); 5Division of Dental Anatomy, Department of Oral Science, Graduate School of Dentistry, Kanagawa Dental University, 82 Inaoka, Yokosuka, Kanagawa 2388580, Japan; m.tou@kdu.ac.jp; 6Department of Highly Advanced Stomatology, Graduate School of Dentistry, Kanagawa Dental University, 3-31-6 Tsuruya, Kanagawa-ku, Yokohama, Kanagawa 2210835, Japan; shimizu@kdu.ac.jp (T.S.); kamata@kdu.ac.jp (Y.K.); 7Division of Histology, Embryology and Neuroanatomy, Department of Oral Science, Graduate School of Dentistry, Kanagawa Dental University, 82 Inaoka, Yokosuka, Kanagawa 2388580, Japan; kawata@kdu.ac.jp

**Keywords:** high-fat, fructo-oligosaccharide, saliva, IgA, sympathetic nerve

## Abstract

Consumption of indigestible dietary fiber increases immunoglobulin A (IgA) levels in saliva. The purpose of this study is to clarify the synergistic effect of the intake of a high amount of fats and indigestible dietary fiber on IgA levels in saliva and submandibular glands (SMG). Seven-week-old Wistar rats were fed a low-fat (60 g/kg) fiberless diet, low-fat fructo-oligosaccharide (FOS, 30 g/kg) diet, high-fat (220 g/kg) fiberless diet, or high-fat FOS diet for 70 days. The IgA flow rate of saliva (IgA FR-saliva) was higher in the low-fat FOS group than in the other groups (*p* < 0.05). Furthermore, the concentration of tyrosine hydroxylase (a marker of sympathetic nerve activation) in the SMG was higher in the low-fat FOS group (*p* < 0.05) and positively correlated with the IgA FR-saliva (rs = 0.68. *p* < 0.0001. *n* = 32) in comparison to that in the other groups. These findings suggest that during low-fat FOS intake, salivary IgA levels may increase through sympathetic nerve activation.

## 1. Introduction

The oral cavity is an entry point for foods, pathogenic microorganisms, and viruses [1]. Therefore, to prevent systemic infections, the oral cavity has developed symbiotic interactions between microorganisms and the host as well as an active innate immune system [2]. The surface of the oral mucosa is mainly protected by saliva that is constantly secreted by major and minor salivary glands [3]. The saliva contains many types of defense protein, of which immunoglobulin A (IgA) is the most important and abundant antibody, and it binds specifically to pathogenic microorganisms and viruses, acting as the first line of defense against infection [4]. Mechanistic details of salivary IgA (saliva-IgA) secretion have been elucidated; salivary gland plasma cells produce IgA, which is transported to the oral cavity by the polymeric immunoglobulin receptor (pIgR) expressed in the basolateral membrane of epithelial cells [5]. Therefore, the expression level of the salivary gland pIgR regulates the levels of saliva-IgA in the oral cavity [6].

Saliva-IgA levels are correlated with the risk of upper-respiratory-tract infections (URTIs); low saliva-IgA levels can increase the frequency and duration of URTIs [7,8]. Children under 6 years of age, immunocompromised adults, and the elderly show a high risk of developing more severe URTIs. Therefore, there is a need to establish strategies to increase saliva-IgA levels in these populations [9,10].

Saliva-IgA levels are stimulated by the autonomic nerves [11], which are affected by short-chain fatty acids (SCFAs) production in the large intestine [12]. Indigestible dietary fiber such as fructo-oligosaccharide (FOS) reach the large intestine and are fermented by intestinal bacteria, resulting in the production of SCFAs [13]. Saliva-IgA is controlled by increased production of SCFAs in the intestine via the autonomic nervous system [14,15]. We have previously shown that ingestion of indigestible dietary fiber such as FOS and polydextrose increases saliva-IgA flow and salivary gland pIgR levels [14,15]. Autonomic nerves are affected by high-fat diets [16]. A high-fat diet has been shown to reduce salivary gland levels of tyrosine hydroxylase (TH), a marker for sympathetic nerve activation [17]. Therefore, the amount of fat in the diet might affect saliva-IgA levels. Controlling saliva-IgA production can decrease the risk of URTIs. McFarlin et al. reported that baker’s yeast beta-glucan intake increased salivary IgA levels in runners post-marathon and this finding correlated with reduced URTIs morbidity [8]. Hence, the chemical composition of indigestible dietary fiber and fats in the diet may be a key factor in regulating URTIs that develop as a result of eating habits. 

Fats are risk factors for obesity and diseases associated with obesity such as diabetes, high blood pressure, and cardiovascular disease [18]. Accordingly, the effects of fats are often confounded by those associated with obesity [19,20]. In this study, we controlled for the body weights of rats, thereby eliminating the effects of obesity, and determined the effects of fat consumption on the production of saliva-IgA. The addition of ω-6 unsaturated fatty acids in diets may induce confounding factors associated with obesity [21,22]. Hence, we used lard, saturated fatty acids. This study tested the effects of FOS addition and the amount of fat intake on saliva-IgA levels and the activity of the autonomic nerves. Our aim was to gain insights towards proposing a better diet chemical composition to decrease the risk of developing URTIs.

## 2. Materials and Methods

### 2.1. Animals

This study employed a 2 × 2 factorial two-way design, using 32 male Wistar rats (7-weeks-old), purchased from Japan CLEA (Tokyo, Japan). The study used 7-weeks-old rats, in part, to obtain data on the risks of high fat intake among healthy individuals, including adolescents. In addition, rats at 7 weeks of age could ingest the essential fatty acids without developing a deficiency by adding lard to their diet at a weight ratio of 60 g/kg. The rats were housed at 22–24 °C under a 12 h light/dark cycle. After acclimation with a non-purified commercial rodent powdered diet (CE-2, Japan CLEA) for 7 days, the rats were randomly assigned to four groups [groups of low-fat FOS diet (LFFOS), high-fat FOS diet (HFFOS), low-fat fiberless diet (LFFL), and high-fat fiberless diet (HFFL); *n* = 8 each]. All rats were weighed daily. The diet of the high-fat groups was adjusted daily such that individual rats in the control and high-fat groups gained the same amount of weight. 

### 2.2. Ethics Statement

All animal experiments were compliant with the principle of the 3Rs. Our experimental protocol was reviewed and approved by the Ethics Committee for Animal Experiments of Kanagawa Dental University (approval-number: 16-016, approval-date: 18 June 2016). This experiment was performed in accordance with the Guidelines for Animal Experimentation of Kanagawa Dental University and Animal Research: Reporting of In Vivo Experiments (ARRIVE) guidelines for reporting animal research.

### 2.3. Diets

Appendix A lists the composition of the four types of powdered diets. The four types of diets were based on the AIN-76 composition. Corn starch and cellulose were changed to sucrose. FOS (Meioligo P^®^, Meiji Food Material Co. Ltd., Tokyo, Japan) was added at 30 g/kg for the LFFOS and HFFOS groups. Fat was added for the LFFL and LFFOS groups at 60 g/kg and for the HFFL and HFFOS groups at 220 g/kg. The four experimental diets were prepared by Japan CLEA. To eliminate the cornstarch and cellulose contained in AIN-76 reaching the cecum, a no-fiber diet base was employed as in our previous research [14]. To eliminate the potential effects of obesity until the end of the feeding period, we controlled weight gain in the high-fat groups. For accurate body weight gain, rats in the low-fat groups had free access to the experimental diet, whereas those in the high-fat groups were fed an adjusted amount of diet such that their weight gain was equal to that of the rats in the control groups. No deficiency was observed in mineral and vitamin intake in the high-fat groups. The duration of the experiment was 70 days. The aim of this study was to examine the effects of high fat intake during a realistic time period for humans; therefore, the fat intake period was shortened to 70 days [23].

### 2.4. Sampling

We anesthetized all rats for sampling between 23:00 and 03:00 after 70 days. Isoflurane was used as anesthesia. Saliva and whole blood were collected under deep anesthesia. The cecal-digesta, cecal-tissues, livers, and submandibular glands (SMG) were excised after euthanasia. Liver and SMG samples were immediately immersed in 4% paraformaldehyde phosphate buffer solution. The tubes containing blood were rapidly inverted six times and placed at 22 °C for 3 h. After that, serum was separated (1200× *g*, 20 min, and 20 °C). Samples, excluding the liver, were weighed rapidly and stored at −80 °C until subsequent analysis.

### 2.5. Saliva Collection

Five minutes after anesthetization with isoflurane, salivary secretion was elicited by intraperitoneal injection of pilocarpine (8 mg/kg body weight). After salivation began, whole saliva was collected by micropipette for 10 min. Since the doses of anesthetic and pilocarpine affect the flow rate of saliva in rats, the amounts administered were adjusted to be the same for all rats. All saliva samples were stored at −80 °C until analysis.

### 2.6. Measurement of Immunoglobulin A (IgA) Concentration

The total IgA concentration in the cecal-digesta, serum, and saliva was measured using a Rat IgA enzyme-linked immunosorbent assay (ELISA) Quantitation Kit (BETHYL LABORATORIES Inc., Montgomery, TX, USA) according to the manufacturer’s instructions. The flow rate of saliva (mL/min) was divided by the weight of saliva (mg), with the sampling time being 10 min, and assuming that the specific density of saliva was 1.00 g/mL. The IgA FR-saliva (µg/10 min) was calculated by multiplying the absolute concentration of IgA (µg/mL) by the saliva flow rate (mL/10 min). Our previous study found that the weight of SMG affected IgA FR-saliva levels [14]. Accordingly, the IgA FR-saliva was divided by the weight of the SMG to remove this influence. 

### 2.7. Measurement of Serum Immunoglobulin G (IgG) Concentration

The immunoglobulin G (IgG) concentration in the serum was measured using a Rat IgG ELISA Kit (Immunology Consultants Laboratory Inc., Lake Oswego, OR, USA) according to the manufacturer’s instructions.

### 2.8. Measurement of Tyrosine Hydroxylase (TH) Concentration in Submandibular Glands (SMG)

The concentration of TH in the SMG was measured using a Rat TH ELISA Kit (Elabscience^®^, Houston, TX, USA) according to the manufacturer’s instructions.

### 2.9. pH in Cecal-Digesta Measurement

The pH of the separated cecal-digesta supernatant was measured as previously described [14].

### 2.10. Gene Expression Analysis (Quantitative Real-Time Polymerase Chain Reaction (RT qPCR)) of Polymeric Immunoglobulin Receptor (pIgR) in the Submandibular Gland (SMG)

Total RNA from the SMG was extracted with a RNeasy Protect Mini Kit (QIAGEN, Hilden, Germany) according to the manufacturer’s instructions. Complementary DNA (cDNA) was reverse-transcribed with ReverTra Ace^®^ qPCR RT Master Mix (TOYOBO CO., LTD, Osaka, Japan) according to the manufacturer’s instructions.

The Step One Plus Real Time PCR system (Applied Biosystems, Thermo Fisher SCIENTIFIC, Waltham, MA, USA) was used for polymerase chain reaction (PCR) amplification of the cDNAs with the THUNDERBIRD^®^ SYBR^®^ qPCR Mix (TOYOBO CO., LTD, Osaka, Japan) and oligonucleotide primer pairs specific for pIgR and glyceraldehyde-3-phosphate dehydrogenase (Gapdh) mRNAs (Eurofins Genomics, Tokyo, Japan). The primer-sequences were as follows: pIgR forward-primer 5’-CAGTCCTCGAAGGAAAAGATGAAAT-3’ and reverse-primer 5’-CAGGAATGCTGAGTAGGCCA-3’; Gapdh forward-primer 5’-GTATCGGACGCCTGGTTAC-3’ and reverse-primer 5’-CTTGCCGTGGGTAGAGTCAT-3’. Quantitative real-time polymerase chain reaction (qRT PCR) was performed using the 2-step program with the following parameters provided by the manufacturer: 1 min at 95 °C, followed by 40 cycles of denaturation at 95 °C for 15 s, and extension at 60 °C for 45 s. At the end of the 40 cycles, melting curve analyses were performed using the following protocol: 15 s at 95 °C, 15 s at 60 °C, and 15 s at 95 °C. Gapdh was used to normalize mRNA levels by applying the ΔΔCt method.

### 2.11. Submandibular Gland (SMG) and Liver Tissue Preparation for Hematoxylin and Eosin Staining

SMG and liver samples were fixed in 4% paraformaldehyde phosphate buffer solution for 24 h. The samples were then embedded in paraffin. Sections (4 μm) were stained with hematoxylin and eosin (H&E). The staining of specimens was recorded using a digital camera for light microscopy (DP 27, Olympus Corporation, Tokyo, Japan).

### 2.12. Statistical Analysis

Statistical analyses were carried out using statistics software (JMP version-12; SAS Institute Japan, Tokyo, Japan). All results are expressed as the group mean ± standard error of the mean (SEM). Data were analyzed using two-way analysis of variance (ANOVA) to show two main effects and the interaction effect. Tukey’s multiple comparison was used for post-hoc analysis when the interaction was significant. Correlations between two variables were analyzed using the Pearson product-moment correlation coefficient. To evaluate the significance of differences in weight gain, diet intake per day, energy intake per day, fat intake per day, and FOS intake per day, one-way ANOVA was performed. As a post-hoc test, differences between groups were analyzed using Tukey’s multiple comparison test after one-way ANOVA. The significance level was defined at *p* < 0.05.

## 3. Results

### 3.1. Rat Weight Gain and Energy, Diet, and Fat Intake

There were no marked differences between the four groups in weight gain over the 70 days nor in energy intake per day (*p* = 0.9 and *p* = 0.6, respectively; one-way ANOVA; Table 1). Differences in dietary intake and fat intake per day were observed among the groups (*p* < 0.0001, <0.0001, respectively; one-way ANOVA; Table 1); low-fat groups had significantly higher diet intake and lower fat intake than those in the high-fat groups (*p* < 0.05 and *p* < 0.05, respectively; Tukey’s multiple comparison).

### 3.2. Fatty Degeneration in Rat Liver Tissue and Submandibular Gland (SMG) Tissue

H&E staining results showed no fatty degeneration-like findings in the rat liver or SMG in all groups after 70 days of treatment (Figure 1).

### 3.3. Effect of Fat Intake and Fructo-Oligosaccharide (FOS) Supplementation

There were no interaction effects observed between fat intake and FOS supplementation or effects of fat intake or FOS supplementation alone on the weight of SMG or flow rate of saliva (interaction: *p* = 0.08, 0.1, respectively; fat intake: *p* = 0.8, 0.07, respectively; FOS supplementation: *p* = 0.7, 1.0, respectively; two-way ANOVA; Figure 2).

Furthermore, no interaction effects were observed between fat intake and FOS supplementation or effects of fat intake alone on the IgA concentration in cecal-digesta and the weight of cecal-digesta (interaction: *p* = 0.1, 0.9, respectively; fat intake: *p* = 0.3, 0.1, respectively; two-way ANOVA; Figure 3). FOS supplementation affected the IgA concentration in cecal-digesta and the weight of cecal-digesta (*p* = 0.02, <0.0001, respectively; two-way ANOVA). The values in the FOS supplementation groups were higher than those in the fiberless groups (Figure 3).

Additionally, no interaction effects were observed between fat intake and FOS supplementation on the weight of cecal-tissue and the pH in cecal-digesta (*p* = 0.2, 0.07, respectively; two-way ANOVA; Figure 4). Fat intake and FOS supplementation affected the weight of cecal-tissue and the pH in cecal-digesta (fat intake: *p* < 0.0001, <0.0001, respectively; FOS supplementation: *p* < 0.0001, <0.0001, respectively; two-way ANOVA). For cecal-tissue, the values in the FOS supplementation groups were higher than those in the fiberless groups. Furthermore, the values in the low-fat groups were higher than those in the high-fat groups (Figure 4). On the contrary, in the cecal-digesta, the pH values in the FOS supplementation groups were lower than those in the fiberless groups and those in the low-fat groups were lower than those in the high-fat groups (Figure 4).

There were no interaction effects between fat intake and FOS supplementation or effects of FOS supplementation alone on the concentration of IgA in serum (interaction: *p* = 1.0; FOS supplementation: *p* = 0.2; two-way ANOVA; Figure 5A). Fat intake affected the concentration of IgA in serum (*p* = 0.01; two-way ANOVA; Figure 5A). The values in the low-fat groups were higher than those in the high-fat groups (Figure 5A). Interactions between fat intake and FOS supplementation existed for the serum concentration of IgG (*p* = 0.01; two-way ANOVA; Figure 5B). The serum IgG concentration in the LFFOS group was higher than that in the other groups (*p* < 0.05; Tukey’s multiple comparison; Figure 5B).

Interactions between fat intake and FOS supplementation existed for the concentration of IgA in saliva, IgA FR-saliva per weight of SMG, pIgR mRNA expression levels in SMG, and TH concentration in SMG (*p* = 0.04, 0.03, 0.04, 0.04, respectively; two-way ANOVA; Figure 6). In all cases, the values in the LFFOS group were higher than those in the other groups (*p* < 0.05 in each case; Tukey’s multiple comparison).

### 3.4. Relationship with IgA Flow Rate of Saliva (IgA FR-saliva)

The IgA FR-saliva per weight of SMG was positively correlated with the weight of the cecal-tissue (*r* = 0.66; *p* < 0.0001), TH concentration in SMG (*r* = 0.61; *p* = 0.0003), pIgR mRNA expression level in the SMG (*r* = 0.61; *p* = 0.0003), concentration of IgG in serum (*r* = 0.54; *p* = 0.002), weight of cecal-digesta (*r* = 0.52; *p* = 0.002), and concentration of IgA in serum (*r* = 0.38; *p* = 0.03), as observed using the Pearson product-moment correlation coefficient (*n* = 32 in all cases; Table 2). The IgA FR-saliva level was negatively correlated with the pH in cecal-digesta (*r* = −0.59; *p* = 0.0005; *n* = 32). In contrast, the IgA FR-saliva level did not correlate with the IgA concentration in cecal-digesta (*r* = 0.16; *p* = 0.4; *n* = 32).

### 3.5. Relationship with Polymeric Immunoglobulin Receptor (pIgR) Expression Level in Submandibular Gland (SMG)

The pIgR mRNA expression level in the SMG was positively correlated with the TH concentration in SMG (*r* = 0.71; *p* < 0.0001), weight of cecal-tissue (r = 0.70; *p* < 0.0001), weight of cecal-digesta (*r* = 0.65; *p* < 0.0001), concentration of IgG in serum (*r* = 0.53; *p* = 0.002), and concentration of IgA in serum (*r* = 0.44; *p* = 0.01) using the Pearson product-moment correlation coefficient (*n* = 32 in all cases; Table 3). The pIgR expression level in the SMG was negatively correlated with the pH in cecal-digesta (*r* = −0.70; *p* < 0.0001; *n* = 32). In contrast, the pIgR expression level in the SMG did not correlate with the IgA concentration in cecal-digesta (*r* = 0.19; *p* = 0.3; *n* = 32).

### 3.6. Relationship with Fat Intake per Day

Fat intake per day negatively correlated with the concentration of IgA in serum (*r* = −0.47; *p* = 0.008), concentration of IgG in serum (*r* = −0.47; *p* = 0.008), and TH concentration in SMG (*r* = −0.41; *p* = 0.02) using the Pearson product-moment correlation coefficient (*n* = 32 in all cases; Table 4). In contrast, fat intake per day did not correlate with the weight of cecal-tissue (*r* = −0.27; *p* = 0.1), IgA FR-saliva per weight of SMG (*r* = −0.27; *p* = 0.1), pIgR expression level in SMG (*r* = −0.23; *p* = 0.2), pH in cecal-digesta (*r* = −0.17; *p* = 0.4), weight of cecal-digesta (*r* = −0.12; *p* = 0.5), and IgA concentration in cecal-digesta (*r* = 0.11; *p* = 0.6; *n* = 32 in all cases).

## 4. Discussion

In this study, we examined the effect of the amount of fat and FOS supplementation on IgA FR-saliva. To eliminate the effects of obesity associated with excessive fat intake in this experiment, feed intake was adjusted so that the weight gain of rats in each group was equal. The amount of fat added for the high-fat group was set at ~40% of the total energy intake, similar to high-fat intake in humans [24]. The amount of fat added for the low-fat group was set at approximately 14% of the total energy ingested in 1965, which is before the increase in blood cholesterol in Japan became a public health challenge [25]. A fat intake of 14% of the total energy is considered the low-fat intake of people currently [25]. Therefore, the results of this study provide suggestions for dietary composition for people, including the younger generation, who are not obese but have a high fat intake.

Fat intake and FOS supplementation affected the IgA FR-saliva, and the low-fat with FOS group exhibited a greater effect than that of the remaining groups. We found that reducing fat intake during FOS intake increased the IgA FR-saliva. When low-fat food is ingested, an increase in the IgA FR-saliva of FOS was observed. To increase the IgA FR-saliva levels and reduce the risk of URTIs, it might be more effective not only to consume indigestible dietary fiber such as FOS but also to reduce the amount of fat intake to an energy ratio of 14%.

In the present study, TH concentration in the SMG in the low-fat with FOS group was higher than that in the other groups. Furthermore, there was a positive correlation between IgA FR-saliva and TH concentration in the SMG and a negative correlation between fat intake per day and TH concentration in the SMG. TH is a rate-determining enzyme for catecholamine (mainly dopamine, norepinephrine, and epinephrine) synthesis and is a useful marker for sympathetic nerve activation [26]. Therefore, reducing fat intake would increase sympathetic nerve activity, resulting in increased IgA FR-saliva. Sympathetic nerves might be involved in the increase and decrease of IgA FR-saliva. High-fat dietary intake has been reported to decrease TH expression in the brains of mice [27]. In addition, a high-fat diet reduces the levels of TH in the SMG of mice [17]. The consumption of a high-fat diet based on beef tallow, which is rich in saturated fatty acids such as lard, decreases the norepinephrine turnover rate and sympathetic nerve activity in skeletal muscle, heart, liver, hypothalamus, and cerebral cortex of rats compared with the consumption of a high-fat diet based on safflower oil, which is rich in polyunsaturated fatty acids [16,28,29]. To increase IgA FR-saliva levels, it might be necessary to reduce the overall fat intake, especially meat fat, which is a fat rich in saturated fatty acids [30], in order to increase sympathetic nerve activity.

In this experiment, the FOS supplementation affected the IgA FR-saliva, but the value of IgA FR-saliva of high-fat with FOS was as low as that of the fiberless groups. The pIgR mRNA expression levels in SMG and TH concentration in SMG showed similar results. Decreased sympathetic nerve activity due to high-fat intake would have reduced SMG pIgR expression and reduced IgA FR-saliva. IgA in saliva is produced as dimeric IgA by salivary gland plasma cells and then transported to the lumen by the pIgR expressed on the basal surface of salivary gland epithelial cells [5]. Therefore, the transport of IgA to saliva depends on the expression level of pIgR on the basal surface of epithelial cells of the salivary glands [31]. In this study, the expression of pIgR in the SMG showed a positive correlation with the IgA FR-saliva. In our previous study, FOS intake increased SCFAs production in the rat cecum, which increased SMG pIgR expression [14,32]. SCFAs produced by bacteria in the gut activate sympathetic nerves via G-protein-coupled receptor 41, a receptor for SCFAs expressed in the sympathetic ganglia [33]. It has been reported that salivary gland pIgR expression is increased by stimulation of both sympathetic nerves and parasympathetic nerves [34], and that the effect may be higher with sympathetic nerve stimulation [35,36]. High-fat intake reduces sympathetic nerve activation. Ke et al. has reported that levels of Bacteroidetes in the feces of mice fed a high-fat diet were reduced as well as SCFAs levels in the cecal-digesta [37]. Although the SCFAs concentration of cecal-digesta was not measured in this experiment, fat intake affected weight of cecal-tissue and pH in cecal-digesta. High-fat intake resulted in low weight of cecal-tissue and high pH in cecal-digesta. Therefore, in this experiment, the SCFAs level in the cecal-digesta would have been decreased due to high-fat intake. The low sympathetic nerve activity of the high-fat with FOS group compared to that of the low-fat with FOS group in this experiment might have been influenced by the decrease in cecal SCFA levels due to high-fat intake. We found that the beneficial effect of the intake of indigestible dietary fiber on IgA FR-saliva levels by increasing SMG pIgR expression was counteracted by fat intake of 40% energy ratio, which is a high amount of fat intake in humans.

Both fat intake and FOS supplementation affected the concentration of IgG in serum; the IgG concentration in the low-fat with FOS group was higher than that in the other groups. The concentration of IgA in serum was affected by fat intake but not FOS supplementation, and was higher in the low-fat intake group than in the high-fat intake group. Furthermore, a negative correlation was observed between daily fat intake and IgG and IgA concentrations in serum, indicators of systemic immune status [38]. In our previous study, indigestible dietary fiber intake such as FOS and polydextrose affected the intestinal tract but not the concentration of IgA in serum [14,15,32]; the same result was obtained in this study. However, it has been reported that FOS intake affects systemic immunity; Csernus et al. reported that ingestion of FOS increased the production level of interferon-γ (IFN-γ) in Peyer’s patch cells, increasing the activity of CD4+ T helper cells, and as a result, the concentration of IgG in serum [39]. Furthermore, FOS intake may have affected serum IgG levels via IFN-γ production in the intestinal tract. 

Previous studies have reported an increased level of serum IgG and IgA in both mice and humans attributed to the effects of obesity associated with a lard-based, high-fat diet [40,41]. In this experiment, the effects of obesity were eliminated; thus, a high-fat diet intake would not have resulted in high serum IgA and IgG levels. Zalcman et al. reported that sympathetic nerve stimulation increased antibody-producing cells in a plaque forming cell assay using sheep red blood cells [42]. In the present study, the SMG concentration of TH, an indicator of sympathetic activation, was higher during low-fat with FOS intake than in the other groups, which would have activated the sympathetic nervous system. Therefore, the increase in antibody-producing cells due to sympathetic activation and the increase in IFN-γ production due to FOS intake might have resulted in higher serum IgG levels, which indicates the state of systemic immunity.

Serum IgG and IgA concentrations were positively correlated with the IgA FR-saliva and the pIgR expression level in the SMG. Stoof et al. reported a strong positive correlation between serum-specific IgG and saliva-specific IgA induced by meningococcal vaccination [43]. The IgA FR-saliva might have been influenced not only by intestinal immunity but also by systemic immunity. High-fat intake might have affected saliva-IgA levels not only from systemic immunity but also from the intestinal tract, and FOS intake might have affected saliva-IgA levels not only from the intestinal tract but also from systemic immunity.

In this experiment, FOS intake affected not only the cecal-tissue and cecal-digesta but also the IgA FR-saliva, which was similarly observed in our previous studies [14,32]. We have previously shown that elevated cecal SCFAs and blood SCFAs levels due to indigestible dietary fiber intake such as FOS would directly affect IgA FR-saliva [14,15]. In this study, the SCFAs levels that increased in the cecum and blood following FOS intake would have increased the IgA FR-saliva.

Our study has some limitations. In many previous studies on high-fat diet intake, not only fat intake, but also obesity caused by a high-fat diet influenced the results of the study [19,20]. In this study, to clarify the effect of fat intake on saliva-IgA levels, the energy intake of rats in the high-fat intake group and the low-fat intake group was equalized to eliminate the effect of obesity. As a result, the high-fat groups had lower dietary and protein intake than the low-fat groups, but the systemic effects of obesity-related inflammation were presumably eliminated. We were unable to measure SCFAs levels in the cecal-digesta and peripheral blood of rats. However, as in previous studies, cecal-digesta weight, cecal-tissue weight, and cecal-digesta pH values represent SCFAs production in the cecum and SCFAs absorption from the cecum into the blood. Because it is impossible to collect spontaneous saliva from rats, we anesthetized the rats during saliva collection and administered pilocarpine intraperitoneally to collect saliva. To ensure a similar effect of anesthetic and pilocarpine administration on rat flow rate of saliva, we administered the same amount of anesthetic and pilocarpine administered to all rats.

## 5. Conclusions

In conclusion, low-fat intake combined with FOS supplementation increased the IgA FR-saliva level, pIgR expression level, and TH concentration in the salivary glands. In addition, high-fat intake reduced the effect of FOS intake on IgA FR-saliva levels. Our results suggest that reducing fat intake increases the IgA FR-saliva levels and that the increase in the IgA FR-saliva levels is associated with sympathetic nerve activation. Furthermore, high-fat intake, such as an energy intake ratio of 40%, might counteract the effects of indigestible dietary fiber intake on increasing saliva-IgA levels. To maximize the beneficial effects of indigestible dietary fiber on saliva-IgA levels, it may be more effective to consume indigestible carbohydrates along with a low-fat diet with an energy intake ratio of 14%, as typically consumed by the Japanese in 1965.

## Figures and Tables

**Figure 1 nutrients-13-01252-f001:**
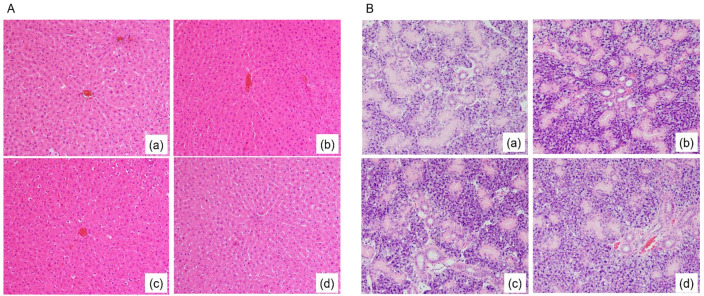
Hematoxylin and eosin staining of liver tissue (**A**) and submandibular gland (SMG) tissue (**B**). The rat groups were fed a low-fat fiberless diet (**a**), low-fat fructo-oligosaccharide diet (**b**), high-fat fiberless diet (**c**), or high-fat fructo-oligosaccharide diet (**d**) for 70 days. Magnification (400×).

**Figure 2 nutrients-13-01252-f002:**
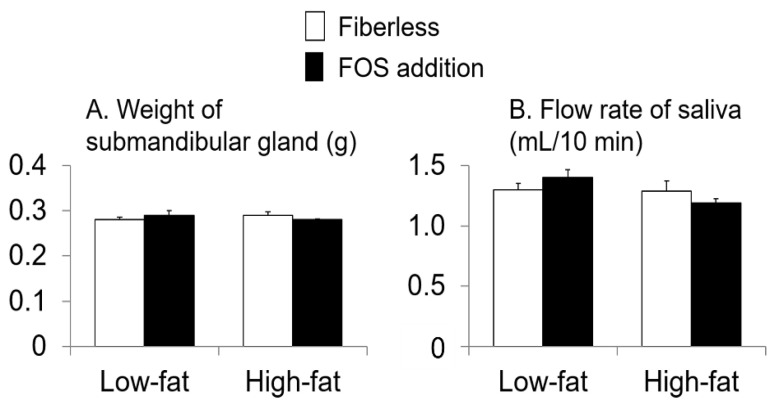
Effects of fat intake and fructo-oligosaccharide (FOS) supplementation on weight of submandibular gland (SMG) (**A**) and flow rate of saliva (**B**). Values are Means ± SEM. *n* = 8 per group. two-way analysis of variance (ANOVA).

**Figure 3 nutrients-13-01252-f003:**
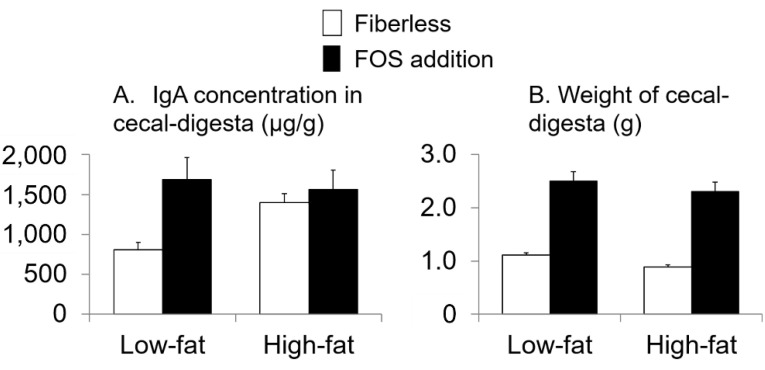
Effects of fat intake and fructo-oligosaccharide (FOS) supplementation on the immunoglobulin A (IgA) concentration in cecal-digesta (**A**) and weight of cecal-digesta (**B**). Values are Means ± SEM. *n* = 8 per group. two-way ANOVA.

**Figure 4 nutrients-13-01252-f004:**
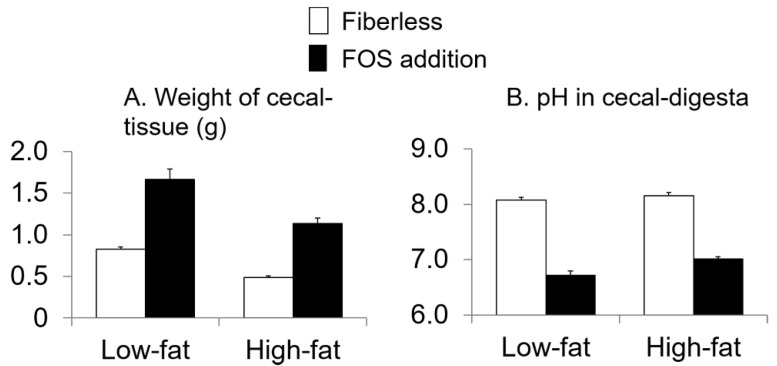
Effects of fat intake and fructo-oligosaccharide (FOS) supplementation on the weight of cecal-tissue (**A**) and pH in cecal-digesta (**B**). Values are Means ± SEM. *n* = 8 per group. two-way ANOVA.

**Figure 5 nutrients-13-01252-f005:**
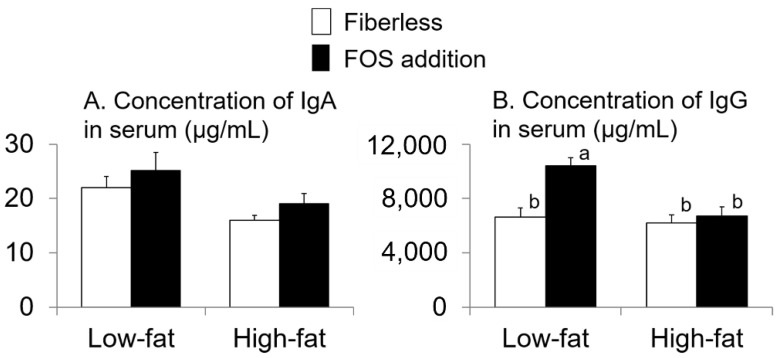
Effects of fat intake and fructo-oligosaccharide (FOS) supplementation on the concentration of IgA (**A**) and immunoglobulin G (IgG) in serum (**B**). Values are Means ± SEM. *n* = 8 per group. two-way ANOVA. Values with different superscript letters are significantly different (*p* < 0.05); Tukey’s multiple comparison.

**Figure 6 nutrients-13-01252-f006:**
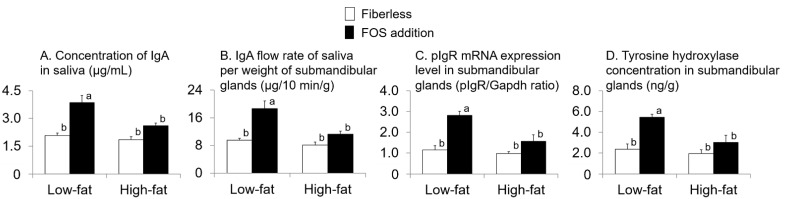
Effects of fat intake and fructo-oligosaccharide (FOS) supplementation on the concentration of IgA in saliva (**A**), IgA flow rate of saliva (IgA FR-saliva) per weight of submandibular glands (SMG) (**B**), polymeric immunoglobulin receptor (pIgR) mRNA expression level in SMG (**C**) and tyrosine hydroxylase (TH) concentration in SMG (**D**). Values are Means ± SEM. *n* = 8 per group. two-way ANOVA. Values with different superscript letters are significantly different (*p* < 0.05); Tukey’s multiple comparison.

**Table 1 nutrients-13-01252-t001:** Rat weight gain over 70 days and per day diet, energy, fat, and fructo-oligosaccharide (FOS) intake.

	LFFL ^‡^	LFFOS ^§^	HFFL *	HFFOS ^#^	*p* ^†^
Weight gain (g/70 days)	178 ± 6	176 ± 6	181 ± 6	179 ± 6	0.9
Diet intake (g/day)	17.5 ± 0.1 ^a^	17.7 ± 0.1 ^a^	14.3 ± 0.0 ^b^	14.6 ± 0.1 ^b^	<0.0001
Energy intake (kJ/day)	292 ± 1	292 ± 2	293 ± 1	294 ± 1	0.6
Fat intake (g/day)	1.05 ± 0.01 ^a^	1.06 ± 0.01 ^a^	3.15 ± 0.01 ^b^	3.18 ± 0.01 ^b^	<0.0001
FOS intake (g/day)	0 ^a^	0.531 ± 0.004 ^b^	0 ^a^	0.438 ± 0.002 ^b^	<0.0001

Mean ± SEM (*n* = 8). Values with different superscript letters are significantly different (*p* < 0.05). LFFL ^‡^: Low-fat fiberless diet. LFFOS ^§^: Low-fat fructo-oligosaccharide diet. HFFL *: High-fat fiberless diet. HFFOS ^#^: High-fat fructo-oligosaccharide diet. ^†^ calculated using one-way ANOVA; post hoc Tukey’s multiple comparison.

**Table 2 nutrients-13-01252-t002:** Correlation between IgA flow rate of saliva (IgA FR-saliva) per weight of submandibular gland (SMG) and different parameters.

	IgA FR-Saliva per Weight of SMG		
	*r* *	*p*	*n*
Weight of cecal-tissue	0.66	<0.0001	32
TH concentration in SMG	0.61	0.0003	32
pIgR expression level in SMG	0.61	0.0003	32
Concentration of IgG in serum	0.54	0.002	32
Weight of cecal-digesta	0.52	0.002	32
Concentration of IgA in serum	0.38	0.03	32
pH in cecal-digesta	−0.59	0.0005	32
IgA concentration in cecal-digesta	0.16	0.4	32

* Pearson product-moment correlation coefficient.

**Table 3 nutrients-13-01252-t003:** Correlation between polymeric immunoglobulin receptor (pIgR) expression level in submandibular gland (SMG) and different parameters.

	pIgR Expression Level in SMG		
	*r* *	*p*	*n*
TH concentration in SMG	0.71	<0.0001	32
Weight of cecal-tissue	0.70	<0.0001	32
Weight of cecal-digesta	0.65	<0.0001	32
Concentration of IgG in serum	0.53	0.002	32
Concentration of IgA in serum	0.44	0.01	32
pH in cecal-digesta	−0.70	<0.0001	32
IgA concentration in cecal-digesta	0.19	0.3	32

* Pearson product-moment correlation coefficient.

**Table 4 nutrients-13-01252-t004:** Correlation between fat intake per day and different parameters.

	Fat Intake per Day		
	*r* *	*p*	*n*
Concentration of IgA in serum	−0.47	0.008	32
Concentration of IgG in serum	−0.47	0.008	32
TH concentration in SMG	−0.41	0.02	32
Weight of cecal-tissue	−0.27	0.1	32
IgA FR-saliva per weight of SMG	−0.27	0.1	32
pIgR expression level in SMG	−0.23	0.2	32
pH in cecal-digesta	−0.17	0.4	32
Weight of cecal-digesta	−0.12	0.5	32
IgA concentration in cecal-digesta	0.11	0.6	32

* Pearson product-moment correlation coefficient.

## Data Availability

Data are available upon request to corresponding author.

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
