# Peer review of "Effect of High Fat and Fructo-Oligosaccharide Consumption on Immunoglobulin A in Saliva and Salivary Glands in Rats"

_nutrients, 2021, doi:10.3390/nu13041252_

Round 1
Reviewer 1 Report
The manuscript “Effect of high fat and fructo-oligosaccharide consumption on immunoglobulin A in saliva and salivary glands in rats” presents results of an interesting study, providing evidences of the relationship between saliva composition and dietary habits.
The manuscript is very well written and very well organized.
Even so, I have some minor points to be addressed.
Introduction
Line 65 – The way authors write the sentence, suggest that URTI can be related with eating habits. A reference supporting this relationship (or a suggestion to this relationship) needs to be added.
Material and methods
Lines 98 – 104 – How did the authors to control weight? Weighed every day? Adjusted the food day-by-day based on weight differences from day-to-day? At individual level, or group level? Please add detail to this part of methodological description.
Line 119 – The authors considered flow rate, but saliva collection was made when animals were under anesthesia and that influence salivation rate. Moreover, in saliva collection section it is not reported the time during which saliva was collected to calculate salivary flow rate.
Results
Line 172 – Please replace “or” by “nor in”
Lines 203 – 205 – From here it appears that higher fiber is the only dietary change that affects IgA concentration in cecal-digesta, but the graph shows that IgA levels were increased also when a fat diet was consumed, even without fiber.
Table 2 – title – I suggest to simplify the title of the table. For example: “Correlation between IgA flow rate of saliva (IgA FR-saliva) per weight of submandibular gland and different parameters”
Discussion
Lines 301-302 – I think authors need to be careful with this statement. Despite the study provides useful information to humans, namely that higher levels of fat have negative effects in the amounts of salivary IgA, to make a relationship between the levels of fat percentage in the diets used and the level of fat percentage in humans. In fact, 14% fat cannot be considered a low fat diet for rats, since their standard diet contains 3-4% fat. So, if getting 14% of energy from fat, in humans is low, in rats it is not.
Lines 369-370 – But this is opposed to what was observed, i.e., if obese ingest higher levels of fat, it would be expected to have lower sIgA levels. These higher levels of sIgA in obese can mean that fat consumed in those studies could be not saturated fat, as in the present study? Can the authors discuss better this part?
Author Response
Response to Reviewer 1’s Comments
We thank for appropriate advice from you very much. The corrected parts are marked with yellow lines in the manuscript.
Introduction
“Line 65 – The way authors write the sentence, suggest that URTI can be related with eating habits. A reference supporting this relationship (or a suggestion to this relationship) needs to be added.”
According to Reviewer 1’s suggestion, we have added a reference to our manuscript that supports the association between URTIs and eating habits (Reference number 18, Line 67-69, Page 2).
Material and methods
“Lines 98 – 104 – How did the authors to control weight? Weighed every day? Adjusted the food day-by-day based on weight differences from day-to-day? At individual level, or group level? Please add detail to this part of methodological description.”
According to Reviewer 1’s suggestion, we have added text on how to manage rats to the manuscript. “All rats were weighed daily. The diet of the high-fat groups was adjusted daily such that individual rats in the control and high-fat groups gained the same amount of weight.” (Line 92-94, Page 2)
“Line 119 – The authors considered flow rate, but saliva collection was made when animals were under anesthesia and that influence salivation rate. Moreover, in saliva collection section it is not reported the time during which saliva was collected to calculate salivary flow rate.”
According to Reviewer 1’s suggestion, we added to the manuscript an explanation of the effects of anesthesia and pilocarpine administration on salivary rate in rats. We also added saliva collection time. “Since the doses of anesthetic and pilocarpine affect the flow rate of saliva in rats, the amounts administered were adjusted to be the same for all rats.” (Line 130-132, Page 3), “Because it is impossible to collect spontaneous saliva from rats, we anesthetized the rats during saliva collection and administered pilocarpine intraperitoneally to collect saliva. To ensure a similar effect of anesthetic and pilocarpine administration on rat flow rate of saliva, we administered the same amount of anesthetic and pilocarpine administered to all rats.” (Line 454-458, Page 11), “After salivation began, whole saliva was collected by micropipette for 10 min.” (Line 129-130, Page 3).
Results
“Line 172 – Please replace “or” by “nor in””
According to Reviewer 1’s suggestion, we changed “or” to “nor in”. (Line 193, Page 4)
“Lines 203 – 205 – From here it appears that higher fiber is the only dietary change that affects IgA concentration in cecal-digesta, but the graph shows that IgA levels were increased also when a fat diet was consumed, even without fiber.”
We thank for appropriate advice from you very much. According to Reviewer 1’s suggestion, we reconfirmed the data of IgA concentration in cecal digesta. The values in the Low-fat fibeless group and the values in the High-fat fiberless group were compared using the Student's t-test. As a result, no significant difference was found in the values of the two groups. In addition, the results of two-way ANOVA did not show any effect of fat intake. Therefore, we determined that the IgA concentration in cecal digesta did not change due to the difference in fat intake (Figure 3A, Page 6).
“Table 2 – title – I suggest to simplify the title of the table. For example: “Correlation between IgA flow rate of saliva (IgA FR-saliva) per weight of submandibular gland and different parameters””
According to Reviewer 1’s suggestion, we changed the title of Table 2 to “Correlation between IgA flow rate of saliva (IgA FR-saliva) per weight of submandibular gland (SMG) and different parameters.” (Line 317-318, Page 8). Similarly, we changed the titles of Table 3 and Table 4. “Correlation between polymeric immunoglobulin receptor (pIgR) expression level in submandibular gland (SMG) and different parameters.” (Line 330-331, Page 9), “Correlation between fat intake per day and different parameters.” (Line 343, Page 9)
Discussion
“Lines 301-302 – I think authors need to be careful with this statement. Despite the study provides useful information to humans, namely that higher levels of fat have negative effects in the amounts of salivary IgA, to make a relationship between the levels of fat percentage in the diets used and the level of fat percentage in humans. In fact, 14% fat cannot be considered a low-fat diet for rats, since their standard diet contains 3-4% fat. So, if getting 14% of energy from fat, in humans is low, in rats it is not.”
We thank for appropriate advice from you very much. AIN-76, which was the basis of the diet for this experiment, is a standard diet for rats and contains 5.0% fat by weight. The fat added to the Low-fat diet in this experiment has a weight ratio of 6.0%, which is not a high-fat diet compared to AIN-76. The Low-fat diet has a 14% fat addition at the calorie ratio, which is a fat intake that falls within the low-fat range in humans. Therefore, we considered the fat-added diet with a weight ratio of 6.0% and a calorie ratio of 14% to be low in fat, and described it as a “low-fat diet” in the manuscript.
“Lines 369-370 – But this is opposed to what was observed, i.e., if obese ingest higher levels of fat, it would be expected to have lower sIgA levels. These higher levels of sIgA in obese can mean that fat consumed in those studies could be not saturated fat, as in the present study? Can the authors discuss better this part?”
According to Reviewer 1’s suggestion, we have added the following text to the manuscript: “Previous studies have reported an increased level of serum IgG and IgA in both mice and humans attributed to the effects of obesity associated with a lard-based, high-fat diet [41,42]. In this experiment, the effects of obesity were eliminated; thus, a high-fat diet in-take would not have resulted in high serum IgA and IgG levels.” (Line 420-423, Page 11)
Reviewer 2 Report
The article entitled “Effect of high fat and fructo-oligosaccharide consumption on 2 Immunoglobulin A in saliva and salivary glands in rats” focuses on the effects of fat and indigestible carbohydrates diets on salivary IgA and related parameters in rat. The subject is very interesting and important since it shows how diet can modulate salivary composition and by this way, the physiological functions of saliva. The experiments are well performed. However, some explanations and corrections have to be done.
L 433: All the references need to be numbered as they are quoted in the text.
L 40: the term “viruses” is too restrictive, it would be more appropriate to replace by “microorganisms” or “microorganisms and viruses”
L 58: Can you explain the mechanisms leading to the production of SCFAs from FOS. I guess, it is fermentation by bacterias?
L 66: the term “non communicable” diseases” is too vague and probably too large. Please use an alternative or give examples.
The last paragraph of the introduction is a bit confusing and could be simplified. I do not think it is important for your study to give such explanation on the importance of the omega-6/omega-3 ratio.
L 79: Why did you choose the age of 7 weeks?
L104: Why did you choose a duration of 70 days?
L 100: Did you check that the restrictive feeding access of the HF groups did not lead to a lack in some nutrients? I am not sure that based the weight gain of the HF group on the weight gain of the other groups was sufficient. Since the added fat is more energetic, it is possible that the HF groups did not consume a sufficient amount in other nutrients and vitamins.
L 115: Please detail more the saliva sampling method. It is important to do it especially if you use pilocarpine. Can you also detail how IgA FR is calculated and why it is interesting to consider this new variable.
L 164: Why did you choose to use Spearman’s rank correlation instead of Pearson?
L263: “submandibular” instead of “subman”
Figures 2 ; 4 ; 5; 6: Please use bar charts with SEM. There is no reason of connecting the values of the different groups. Please use marks such as stars or crosses when the results are significant (e.g. fig 3 ; fig 4)
Global to discussion: Please do not quote the figures and tables in the discussion; this should be kept for the result section. This makes this part quite hard to read. This part can also be shortened since many sentences are repetitions of results and some of them express the same idea (e.g: l 346-347). Please consider revising this part to make it more easy to read.
L 295: The term “flow rate” here can be confusing with saliva flow rate. Please modify to avoid such misinterpretation. Taking about IgA level is sufficient to me.
L 302: “who are high in fat”: This sentence is too vague. Are you talking about overweight, blood cholesterol or something else?
Author Response
Response to Reviewer 2’s Comments
We thank for appropriate advice from you very much. The corrected parts are marked with yellow lines in the manuscript.
“L 433: All the references need to be numbered as they are quoted in the text.”
I'm sorry that the reference numbers have disappeared. We have included the reference number in the manuscript (Line 490-588, Page 13-14).
“L 40: the term “viruses” is too restrictive, it would be more appropriate to replace by “microorganisms” or “microorganisms and viruses””
According to Reviewer 2’s suggestion, we changed the “virus” to “microorganisms” (Line 42, Page 1).
“L 58: Can you explain the mechanisms leading to the production of SCFAs from FOS. I guess, it is fermentation by bacterias?”
According to Reviewer 2’s suggestion, we have added the following text to the manuscript: “Indigestible dietary fiber such as fructo-oligosaccharide (FOS) reach the large intestine and are fermented by intestinal bacteria, resulting in the production of SCFAs [13].” (Line 58-60, Page 2).
“L 66: the term “non communicable” diseases” is too vague and probably too large. Please use an alternative or give examples.”
According to Reviewer 2’s suggestion, we changed the term “non communicable” to concrete term. “diseases associated with obesity such as diabetes, high blood pressure, and cardiovascular disease” (Line 72-73, Page 2).
“The last paragraph of the introduction is a bit confusing and could be simplified. I do not think it is important for your study to give such explanation on the importance of the omega-6/omega-3 ratio.”
According to Reviewer 2’s suggestion, we have removed the Omega 3 / Omega 6 ratio description from the manuscript to simplify the text (Line 76-78, Page 2).
“L 79: Why did you choose the age of 7 weeks?”
According to Reviewer 2’s suggestion, we added to the manuscript why we used 7-weeks- old rats. “The study used 7-weeks-old rats, in part, to obtain data on the risks of high fat intake among healthy individuals, including adolescents. In addition, rats at 7 weeks of age could ingest the essential fatty acids without developing a deficiency by adding lard to their diet at a weight ratio of 60 g/kg.” (Line 85-88, Page 2)
“L104: Why did you choose a duration of 70 days?”
According to Reviewer 2’s suggestion, we added to the manuscript the reason why the rats breeding period was 70 days. “The duration of the experiment was 70 days. The aim of this study was to examine the effects of high fat intake during a realistic time period for humans; therefore, the fat intake period was shortened to 10 weeks [24].” (Line 114-117, Page 3)
“L 100: Did you check that the restrictive feeding access of the HF groups did not lead to a lack in some nutrients? I am not sure that based the weight gain of the HF group on the weight gain of the other groups was sufficient. Since the added fat is more energetic, it is possible that the HF groups did not consume a sufficient amount in other nutrients and vitamins.”
We confirmed in this experiment that all rats were not deficient in vitamin and mineral intake as well as essential fatty acids. According to Reviewer 2’s suggestion, we have added the following text to the manuscript: “No deficiency was observed in mineral and vitamin intake in the high-fat groups.” (Line 114-115, Page 3)
“L 115: Please detail more the saliva sampling method. It is important to do it especially if you use pilocarpine. Can you also detail how IgA FR is calculated and why it is interesting to consider this new variable.”
According to Reviewer 2’s suggestion, we have added detailed saliva collection methods and IgA flow rate of saliva calculation methods to the manuscript. “Five minutes after anesthetization with isoflurane, salivary secretion was elicited by intraperitoneal injection of pilocarpine (8 mg/kg body weight). After salivation began, whole saliva was collected by micropipette for 10 min. Since the doses of anesthetic and pilocarpine affect the flow rate of saliva in rats, the amounts administered were adjusted to be the same for all rats.” (Line 128-132, Page 3), “The flow rate of saliva (mL/min) was divided by the weight of saliva (mg), with the sam-pling time being 10 min, and assuming that the specific density of saliva was 1.00 g/mL. The IgA FR-saliva (µg/10 min) was calculated by multiplying the absolute concentration of IgA (µg/mL) by the saliva flow rate (mL/10 min). Our previous study found that the weight of SMG affected IgA FR-saliva levels [14]. Accordingly, the IgA FR-saliva was di-vided by the weight of the SMG to remove this influence.” (Line 136-142, Page 3)
“L 164: Why did you choose to use Spearman’s rank correlation instead of Pearson?”
In our current study, we were able to confirm the homoscedasticity of the data. Therefore, according to Reviewer 2’s suggestion, we retested Table 2, Table 3, and Table 4 with Pearson product-moment correlation coefficient. The results were similar to those tested by Spearman's rank correlation coefficient, so the content of the discussion was not affected (Table 2, Table 3 and Table 4, Page 8-9).
“L263: “submandibular” instead of “subman””
Thank you for pointing out the misspelling. We have fixed it to the correct "submandibular" (Line 320, Page 8).
“Figures 2 ; 4 ; 5; 6: Please use bar charts with SEM. There is no reason of connecting the values of the different groups. Please use marks such as stars or crosses when the results are significant (e.g. fig 3 ; fig 4)”
According to Reviewer 2’s suggestion, the graphs of two-way ANOVA have been changed to bar graphs. If no interaction was observed in the two-way ANOVA, subsequent multiple comparisons could not be performed (Figure 2, 3, 4, 5A). However, those with interactions were subsequently subjected to multiple comparisons, and the data with significant differences were written in different alphabets (Figure 5B, 6).
“Global to discussion: Please do not quote the figures and tables in the discussion; this should be kept for the result section. This makes this part quite hard to read. This part can also be shortened since many sentences are repetitions of results and some of them express the same idea (e.g: l 346-347). Please consider revising this part to make it more easy to read.”
According to Reviewer 2’s suggestion, we have removed all figure and table citations from “Discussion”. I also removed the resulting citation for simplification.
“L 295: The term “flow rate” here can be confusing with saliva flow rate. Please modify to avoid such misinterpretation. Taking about IgA level is sufficient to me.”
According to Reviewer 2’s suggestion, we changed “flow rate” to “IgA FR-saliva” (Line 347, Page 9).
“L 302: “who are high in fat”: This sentence is too vague. Are you talking about overweight, blood cholesterol or something else?”
According to Reviewer 2’s suggestion, we changed “who are high in fat” to “who are not obese but have a high fat intake” (Line 355, Page 9).